# Controllable Blind AC FDIA via Physics-Informed Extrapolative AVAE

**DOI:** 10.3390/s25030943

**Published:** 2025-02-05

**Authors:** Siliang Zhao, Wuman Luo, Qin Shu, Fangwei Xu

**Affiliations:** 1College of Electrical Engineering, Sichuan University, Chengdu 610000, China; 15913109663@163.com (S.Z.); shuqin@scu.edu.cn (Q.S.); 2School of Applied Sciences, Macao Polytechnic University, Macao, China; luowuman@mpu.edu.mo

**Keywords:** controllable false data injection attack, AC state estimation, physics informed, data driven, extrapolative adversarial variational autoencoder

## Abstract

False data injection attacks (FDIAs) targeting AC state estimation pose significant challenges, especially when only power measurements are available, and voltage measurements are absent. Current machine learning-based approaches struggle to effectively control state estimation errors and are confined to the data distribution of training sets. To address these limitations, we propose the physics-informed extrapolative adversarial variational autoencoder (PI-ExAVAE) for generating controllable and stealthy false data injections. By incorporating physically consistent priors derived from the AC power flow equations, which enforce both the physical laws of power systems and the stealth requirements to evade bad data detection mechanisms, the model learns to generate attack vectors that are physically plausible and stealthy while inducing significant and controllable deviations in state estimation. Experimental results on IEEE-14 and IEEE-118 systems show that the model achieves a 90% success rate in bypassing detection tests for most attack configurations and outperforms methods like SAGAN by generating smoother, more realistic deviations. Furthermore, the use of physical priors enables the model to extrapolate beyond the training data distribution, effectively targeting unseen operational scenarios. This highlights the importance of integrating physics knowledge into data-driven approaches to enhance adaptability and robustness against evolving detection mechanisms.

## 1. Introduction

Power systems constitute critical social infrastructure, making safety a paramount consideration in the operation of modern electrical power systems. Recently, the widespread adoption of information technology and the deep integration of physical and cyber domains have posed significant challenges for cybersecurity in contemporary power systems.

The concept of stealthy false data injection attacks (FDIAa) targeting power system state estimation (SE) has garnered considerable attention in the field, as discussed in [1]. These attacks are notable for their capacity to manipulate system data undetected, thereby posing serious risks to the integrity of state estimation processes. Given that state estimation is vital for various power system operations, including economic dispatch (ED) and contingency analysis [2,3], compromised state estimation can result in erroneous operational and control decisions within the energy management system (EMS). Therefore, it is essential to understand the nature of these cyber threats, implement effective detection mechanisms, and develop robust mitigation strategies.

Research on false data injection attacks (FDIAs) has attracted considerable attention, particularly due to their ability to compromise power system state estimation (SE) with minimal risk of detection [4]. Early explorations into FDIA predominantly focused on DC-based state estimation models. A seminal work by Liu et al. [1] established that if the state estimation Jacobian matrix (SEJM) and system parameters are known, attackers can execute fully unobservable FDIA, while bypassing conventional bad data detection (BDD). Leveraging this foundation, subsequent studies introduced local FDIA [5,6,7,8], where partial system parameters adjacent to the targeted lines suffice. Additionally, blind FDIA, such as DC-based blind methods [9,10,11,12], refer to attacks conducted without direct knowledge of the system’s parameters or topology. Instead, these DC-based approaches rely on techniques like matrix subspace learning to infer necessary information from observable data. Nevertheless, DC-based FDIA methods often prove vulnerable when confronted with AC-based detection techniques in real-world implementations, as illustrated by Rahman et al. [13].

To overcome the limitations associated with DC approximations, researchers have shifted toward constructing FDIA under AC-based state estimation [13]. In this context, Liang et al. [14] found that ensuring complete stealth typically requires knowledge of both the SEJM and certain state variables. Liu et al. [15] proposed a network parameter coordinated false data injection (NP-FDI) attack that reduces the number of attacked measurements by jointly modifying network parameters and power measurements, but it relies on the ability to alter network parameters, which is challenging in practice. Deng et al. [16] introduced a practical FDIA model leveraging power flow or injection measurements to approximate system states; however, it requires knowledge of the admittance matrix and network topology to construct effective attack vectors. Similarly, methods by Zhao et al. [17] and Tian et al. [18] assume access to the system topology and system parameters, making them dependent on accurate system information.

In order to further improve the adaptability of attack methods in practice, more recent research has turned to blind AC FDIAs, which leverage historical measurement data to eliminate the need for prior knowledge of system parameters, such as admittance matrices or topology. For example, the reference [19] introduced a parameter-free FDIA strategy that relies on robust tensor decomposition to construct stealthy attack vectors. However, most existing blind AC FDIA strategies [20,21,22] do not explicitly control the magnitude of the resulting state estimation errors, which becomes critical when attackers aim to engineer precise disruptions or implement adaptive attack strategies. Although a few studies [23,24] consider targeted attacks on state estimation, they all require the introduction of additional state measurements (i.e., voltage measurements).

To address this limitations, this paper introduces the physics-informed extrapolative adversarial variational autoencoder (PI-ExAVAE)-a novel generative model that integrates physical priors from AC power flow equations and adversarial training [25]. In contrast to traditional FDIA methods, PI-ExAVAE enables the following:**Integration of physics-based priors**. PI-ExAVAE incorporates AC power flow equations into the adversarial variational autoencoder framework, ensuring that generated attack vectors are physically consistent and stealthy against traditional detection mechanisms.**Generative extrapolation capability**. Unlike traditional generative models that focus on replicating the training data distribution and generating samples similar to the original input, the PI-ExAVAE integrates physics-informed priors, thereby enabling generative extrapolation to produce physically consistent and stealthy attack vectors far from the original input, even beyond the range covered by the training data.**Precise control over state estimation errors without voltage measurements**. The proposed approach allows fine-grained control over the magnitude and direction of state estimation errors, enabling the design of targeted attacks with predictable impacts.

Experimental results validate the proposed approach, showing that PI-ExAVAE not only maintains stealthiness under AC-based detection but also offers flexible control over the magnitude of state deviation—all without requiring access to historical state information. Our contributions bridge a crucial gap in FDIA research, paving the way for more powerful and adaptive cyberattack and defense frameworks in future power systems.

## 2. Related Work

Recent research has focused on AC-based blind FDIA methods that eliminate reliance on prior knowledge of system parameters, addressing practical limitations of traditional AC-based non-blind FDIA mechanisms. For example, Jiao et al. [20] employed a self-attention-GAN-based (SAGAN) technique. However, this method may lack flexibility for real-time adjustments. Costilla et al. [21] employed a Wasserstein generative adversarial network (WGAN) combined with an autoencoder (AE), which served as a surrogate for the state estimator and regularized the WGAN to produce measurement-consistent attacks capable of bypassing residual-based detectors. However, the AE constrained the attack’s flexibility by limiting the generated attacks to conform closely to the training data distribution, reducing its adaptability to dynamic or unseen scenarios. Furthermore, Narang et al. [22] proposed a LSTMAE-GAN-based FDIA on AC power system state estimation. Their method leverages generative adversarial networks (GANs) to learn measurement data distributions and uses a long short-term memory autoencoder (LSTMAE) as a state estimator mimic to embed physical system laws and temporal dependencies into the generated attack data, achieving high stealthiness and effectiveness without requiring system topology or parameter knowledge. However, the reliance on historical voltage measurements and the predefined temporal structure in LSTMAE limits its adaptability to dynamic or unseen scenarios. Additionally, the LSTMAE architecture primarily focuses on reconstructing temporal sequences rather than exploring broader data distributions, which restricts its ability to generalize to conditions outside the training data. In contrast, our PI-ExAVAE leverages the latent space of a variational autoencoder (VAE), which provides a more flexible representation of the system’s measurement-to-state mapping. By structuring the latent space with physical priors and incorporating adversarial training, PI-ExAVAE extends beyond the training data distribution while ensuring physical consistency. Unlike LSTMAE, which is inherently sequence-dependent, VAE’s latent space supports both static and dynamic attack scenarios, enabling precise control over state deviations without relying on temporal patterns or voltage measurements. Afrin and Ardakanian [26] proposed Sneaky-FGSM, a practical attack method using surrogate models trained on high-quality historical data, including voltage measurements. While effective at bypassing detection mechanisms, its reliance on surrogate models trained on historical voltage measurements reduces its adaptability in scenarios where such data are unavailable or outdated. This limitation makes it challenging to achieve coordinated and consistent adjustments across all state variables, particularly in high-dimensional systems. While these methods achieve blind attacks on AC state estimation using only power measurements, they fail to provide precise control over post-attack state deviations. This restricts their capacity to achieve desired impacts.

Targeted FDIA techniques have been proposed to achieve precise control over state estimation deviations by constructing stealthy attack vectors. Chin et al. [27] introduced a blind FDIA using a geometric approach to construct stealthy vectors by minimizing the angle between measurement and attack vectors. A targeted variant was also proposed, enabling precise control over specific state variables but requiring additional voltage information for implementation. Du et al. [28] proposed a targeted FDIA model for AC state estimation that bypasses the need for network parameters by leveraging limited PMU data. The Ornstein–Uhlenbeck process and regression theorem [29] can estimate key parameters like line admittances to construct stealthy attack vectors inducing large deviations in state variables. However, the effectiveness depends on accurate PMU voltage measurements for parameter estimation and attack design. Rahman et al. [30] proposed an adversarial model that leverages artificial neural networks (ANNs) to infer grid topology from historical measurements. However, the method relies on the availability of voltage measurements to accurately construct a substitute topology, which is then integrated into the attack pipeline along with a substitute bad data detector. The proposed method in [31] advances FDIA design by using adversarial machine learning to improve stealthiness against BDD and NAD, while sparse-state attacks enhance scalability and reduce costs. However, its reliance on a white-box scenario limits practicality. All of the above methods enable some degree of control over state estimation deviations by establishing a mapping between power measurements and state variables (e.g., voltage). However, their ability to achieve control depends on the availability of voltage measurements in the observation data, which are essential for constructing the mapping. In practice, such data may not always be accessible, especially in systems with only power measurements. This limitation underscores the need for an approach capable of precisely controlling state estimation deviations without relying on voltage measurements.

Our PI-ExAVAE addresses the limitations of existing methods, particularly their reliance on voltage measurements, by leveraging the controllability of the VAE’s latent space to construct false data injection attacks. Inspired by [27] and the aforementioned GAN-based works, we structure the latent space with physical priors and integrate adversarial training to achieve controllable and stealthy attacks. Unlike Sneaky-FGSM, LSTMAE-GAN, and AE-WGAN, our method does not require historical voltage measurements, which broadens its applicability in systems lacking comprehensive state observations. Moreover, PI-ExAVAE provides precise control over the magnitude of deviations across all state variables, enabling coordinated and consistent adjustments that surpass the capabilities of existing methods such as SA-GAN, which focuses on generating stealthy attack vectors but lacks the ability to control specific or global state deviations. Table 1 summarizes the key differences between our method and previous works, highlighting PI-ExVAE’s ability to control post-attack state estimation deviations while eliminating reliance on voltage measurements.

## 3. Fundamentals of False Data Injection Attacks on AC State Estimation

In contemporary power systems, the weighted least squares (WLSs) estimation method, detailed in Equation (Equation 1), plays a crucial role in estimating system states.(1)$$\hat{x}=\underset{x}{\arg\min}{\left[y-h\left(x\right)\right]}^{T}R^{-1}\left[y-h\left(x\right)\right]$$
where x represents the system state, while y denotes measurement data, including bus injections and bidirectional line flows. The estimated system state, $\hat{x}$ is derived to best fit the observed measurements y. The relationship between measurements and system states is defined by the nonlinear function *h* representing the nonlinear function between measurement data and system state, which depends on network topology and line parameters [32]. R is a diagonal matrix expressed as $R=diag(\sigma_{1}^{2},\cdot \cdot \cdot \sigma_{i}^{2},\cdot \cdot \cdot ,\sigma_{m}^{2})$, where $\sigma_{i}^{2}$ is the variance of the measurement error associated with the *i*-th meter and *m* is the number of measurements. Measurement errors and disturbances can result in y≠h(x). To address this, system operators employ bad data detection techniques to identify and filter out anomalies. These methods include the Chi-squares $\chi^{2}$-test, as well as more precise approaches utilizing normalized residuals [33]. The latter involves a series of well-defined steps to enhance detection accuracy:Step 1. Solve the WLS estimation and obtain the elements of the measurement residual vector:(2)$$r_{i}=y_{i}-h_{i}\left(\hat{x}\right)$$Step 2. Compute the normalized residuals:(3)$$\begin{array}{cc} r_{i}^{N} & =\frac{|r_{i}|}{\sqrt{R_{ii}S_{ii}}}=\frac{|r_{i}|}{\sqrt{R_{ii}S_{ii}}}\\ S & =I-HG^{-1}H^{T}R^{-1}\\ G & =H^{T}R^{-1}H \end{array}$$
where H is the Jacobian of the linearized system dynamics, which is determined by the power network topology and the admittances of the branches.Step 3. Find *k* such that $r_{k}^{N}$ is the largest among all $r_{i}^{N}$.Step 4. If $r_{k}^{N}>\tau$, then the k−th measurement will be suspected bad data. Here, τ is a chosen identification threshold, for instance 3.

In this paper, we test the performance of the proposed model against these two detection methods.

According to the above mentioned processes of AC state estimation, a successful FDIA targeting AC state estimation allows false data to evade bad data detection and appear as legitimate measurements. Let a denote the attack vector and y be the original measurement data. Under attack, the compromised measurements become $y_{a}=y+a$. The resulting residual, $r_{a}$, after the attack is computed as:(4)$$\begin{array}{cc} r_{a} & =∥y_{a}-h\left({\hat{x}}_{a}\right){∥}_{2}\\ \; & =∥y_{a}-h\left({\hat{x}}_{a}\right)+h\left(\hat{x}\right)-h\left(\hat{x}\right){∥}_{2}\\ \; & =∥y-h\left(\hat{x}\right)+a-h\left({\hat{x}}_{a}\right)+h\left(\hat{x}\right){∥}_{2} \end{array}$$
so, the attack vector a should satisfy:(5)$$a=h\left({\hat{x}}_{a}\right)-h\left(\hat{x}\right)$$
and the key to designing a false data injection attack on the AC state estimation lies in identifying the nonlinear function h that connects measurement data with the power system state.

## 4. Proposed AC FDIA Method Based on PI-ExAVAE

In this section, the physics-informed extrapolative adversarial variational autoencoder (PI-ExAVAE) framework is designed to generate physically consistent, stealthy, and controllable FDIA targeting AC state estimation. Unlike conventional generative models, PI-ExAVAE incorporates physical priors derived from AC power flow equations into a adversarial variational autoencoder structure, ensuring both extrapolation capability and precise control over state estimation deviations. Below, we detail the components and training methodology of the proposed approach.

### 4.1. Variational Autoencoder with Adversarial Loss

A variational autoencoder (VAE) is composed of two neural networks: an encoder and a decoder. The encoder encodes a measurement sample y into a latent representation z, while the decoder reconstructs the original data $\hat{y}$ from z. This process can be described as:(6)$$\begin{array}{cc} z & ∼q\left(z|y\right)=N\left(\mu_{z},\sigma_{z}^{2}\right),\;\left(\mu_{z},\sigma_{z}\right)=\mathrm{Enc}\left(y\right)\\ \hat{y} & ∼p\left(y|z\right)=N\left(\mu_{y},\sigma_{y}^{2}\right),\;\left(\mu_{y},\sigma_{y}\right)=\mathrm{Dec}\left(z\right) \end{array}$$
where Enc(·) and Dec(·) represent the mapping functions of the encoder and decoder, respectively. To ensure meaningful latent representations, the VAE imposes a prior p(z), typically chosen as z∼N(0,I). The VAE loss is minus the sum of the expected log likelihood (the reconstruction error) and a prior regularization term [34]:(7)$$L_{\mathrm{VAE}}=-E_{q(z|y)}\left[\log\frac{p(y|z)p(z)}{q(z|y)}\right]=L_{\mathrm{llike}}+L_{\mathrm{prior}}$$
with(8a)$$\begin{array}{cc} L_{\mathrm{llike}} & =-E_{q(z|y)}\left[\log p(y|z)\right] \end{array}$$(8b)$$\begin{array}{cc} L_{\mathrm{prior}} & =D_{KL}\left(q(z|y)||p(z)\right) \end{array}$$
where $D_{KL}$ is the Kullback–Leibler divergence and the subscript llike represents the log-likelihood function. Furthermore, to enhance the realism and stealth of generated samples, an adversarial discriminator D(y) is introduced. This discriminator is a neural network that attempts to differentiate between real measurement samples y and reconstructed/generated samples $\hat{y}$. The adversarial component is trained jointly with the VAE to improve the quality of reconstructions and ensure they resemble real-world measurements. The adversarial loss is defined as:(9)$$L_{\mathrm{adv}}=E_{y∼p_{\mathrm{data}}}\left[\log D(y)\right]+E_{\hat{y}∼p_{\mathrm{vae}}}\left[\log\left(1-D(\hat{y})\right)\right]$$
where $p_{\mathrm{data}}$ and $p_{\mathrm{vae}}$ are the true distribution of measurements and distributions of reconstructed samples from the VAE, respectively.

### 4.2. Proposed Physics-Informed PI-ExAVAE

In the analysis presented in Section 3, the function *h* plays a critical role in constructing the attack vector. However, calculating *h* heavily relies on network information, which is typically assumed to be inaccessible to potential attackers. On the other hand, the adversarial VAE, as introduced in Section 4.1, is primarily designed to improve the reconstruction of original data. This objective differs from our goal of generating fake data for specific purposes.

To address this challenge, ref. [20] proposes using measurements from a different time as a reference to generate false measurements. While this method can effectively bypass BDD detection, it lacks control over the bias introduced into state estimation by the false measurements. To ensure control over state deviation and expand the generation space of fake data within the constraints of physical laws, we propose a novel physical loss function inspired by [27,35]. Based on Equation (Equation 1), the state deviation caused by an attack near the operating point can be expressed as follows:(10)$$\Delta x={\left(H^{T}R^{-1}H\right)}^{-1}H^{T}R^{-1}\left(h\left({\hat{x}}_{a}\right)-h\left(\hat{x}\right)\right)$$
then, substituting Equation (Equation 5) into Equation (Equation 10), the relationship between the attack vector a and Δx can be obtained as a=HΔx. The active and reactive power injections at bus *i* measurements and power flow measurements from bus *i* to bus *j* can be expressed as follows:(11a)$$\begin{array}{cc} P_{i} & =\sum_{j=1}^{N}V_{i}V_{j}\left(G_{ij}\cos\theta_{ij}+B_{ij}\sin\theta_{ij}\right) \end{array}$$(11b)$$\begin{array}{cc} Q_{i} & =\sum_{j=1}^{N}V_{i}V_{j}\left(G_{ij}\sin\theta_{ij}-B_{ij}\cos\theta_{ij}\right) \end{array}$$(11c)$$\begin{array}{cc} P_{ij} & =V_{i}^{2}G_{ij}-V_{i}V_{j}G_{ij}\cos\theta_{ij}-V_{i}V_{j}B_{ij}\sin\theta_{ij} \end{array}$$(11d)$$\begin{array}{cc} Q_{ij} & =-V_{i}^{2}B_{ij}+V_{i}V_{j}B_{ij}\cos\theta_{ij}-V_{i}V_{j}G_{ij}\sin\theta_{ij} \end{array}$$
where *N* is the number of buses, $\theta_{ij}=\theta_{i}-\theta_{j}$, and $G_{ij}$ and $B_{ij}$ denote the conductance and susceptance, respectively. From Equations (11a)–(11d), since the active and reactive power injections at a bus are equal to the sum of the active and reactive power flows on the connected lines, the sufficient conditions for ‘invisible’ attacks on bus power injections are the same as those for ‘invisible’ attacks on line power flows. Since active power flow and injection are significantly more sensitive to voltage angle than to voltage amplitude, priority is given to sufficient conditions for the attack vector with respect to the phase angle. According to a=HΔx, the relationship between the *m*-th element of a on the active power flow from bus *i* to bus *j* with respect to the phase angle can be expressed as [27]:(12)$$\begin{array}{cc} a_{m} & =\sum_{k=1}^{N}\Delta \theta_{k}\frac{\partial P_{ik}}{\partial \theta_{k}}\\ \; & =\Delta \theta_{ij}V_{i}V_{j}\left(G_{ij}\sin\theta_{ij}-B_{ij}\cos\theta_{ij}\right) \end{array}$$
where $\frac{\partial P_{ik}}{\partial \theta_{k}}=0$ for k≠i,j, and $\Delta \theta_{ij}=\Delta \theta_{i}-\Delta \theta_{j}$, which is the difference of phase angle deviation under attacks. According to Equation (Equation 5), the $a_{m}$ should satisfy the following:(13)$$a_{m}=h_{m}\left({\hat{x}}_{a}\right)-h_{m}\left(\hat{x}\right)=h_{m}\left(\hat{x}+\Delta x\right)-h_{m}\left(\hat{x}\right)\equiv \Delta h_{m}$$
then, according to Equation (11c) and Equation (Equation 13),(14)Δhm=hm(x^+Δx)−hm(x^)=−ViVjGijcos(θ^ij+Δθij)−ViVjBijsin(θ^ij+Δθij)+Vi2Gij−−−−ViVjGijcos(θ^ij)−ViVjBijsin(θ^ij)+Vi2Gij=−ViVjGijcos(θ^ij+Δθij)−ViVjBijsin(θ^ij+Δθij)−+ViVjGijcos(θ^ij)+ViVjBijsin(θ^ij)
next, comparing Equation (Equation 14) with Equation (Equation 13), we aim to reduce $\Delta h_{m}$ to $a_{m}$. Applying the approximations $\cos(\Delta \theta_{ij})\approx 1$ and $\sin\left(\Delta \theta_{ij}\right)\approx \Delta \theta_{ij}$, that is, $\Delta \theta_{ij}\approx 0$, we have the following:(15)−ViVjGijcos(θ^ij+Δθij)−ViVjBijsin(θ^ij+Δθij)+ViVjGijcos(θ^ij)+ViVjBijsin(θ^ij)=−ViVjGijcos(θ^ij)cos(Δθij)−sin(θ^ij)sin(Δθij)−cos(θ^ij)−−ViVjBijsin(θ^ij)cos(Δθij)+cos(θ^ij)sin(Δθij)−sin(θ^ij)≈ΔθijViVjGijsinθij−Bijcosθij=am

Then, because of $\left|\Delta \theta_{i}-\Delta \theta_{j}\right|\leq \left|\Delta \theta_{i}\right|+\left|\Delta \theta_{j}\right|$, by minimizing $\Delta \theta_{i}$ and $\Delta \theta_{j}$, $\Delta \theta_{ij}=\Delta \theta_{i}-\Delta \theta_{j}\approx 0$ can be satisfied. A special case is $\Delta \theta_{i}=0$, which is equivalent to(16)$${\bar{y}}^{T}\bar{a}=1$$
where $\bar{y}=\frac{y}{∥y∥}$ and $\bar{a}=\frac{a}{∥a∥}$ represent the normalized forms of the measurement vector y and the attack vector a, respectively. Then, the VAE reconstruction error of Equation ([Disp-formula FD8a-sensors-25-00943]) can be replaced with(17)$$L_{\mathrm{phy}}={\bar{y}}^{T}\bar{a}-1$$
by replacing reconstruction loss $L_{\mathrm{llike}}$ with physical loss $L_{\mathrm{phy}}$, the proposed PI-ExAVAE leverages the underlying physics of the power system to expand the latent space for generating attack vectors. This structured latent space not only aligns with the system’s operational constraints but also enables the model to extrapolate beyond the training data distribution, generating physically consistent attack vectors under previously unseen scenarios. Moreover, the degree of influence on the system state can be precisely controlled through targeted sampling strategies in the structured latent space. The term ‘targeted sampling strategies’ refers to methods that guide the sampling process in the structured latent space towards specific goals, such as balancing stealth and impact, ensuring physical consistency, or adapting to unseen scenarios. These strategies exploit the structured latent space of PI-ExAVAE to produce attack vectors that achieve their intended objectives, such as causing significant deviations in system states, while remaining physically valid and consistent with power system constraints. This distinct capability to generate diverse attack vectors, adapt to unseen system conditions, and precisely control their impact lies in the ability of PI-ExAVAE to adjust the degree of influence on state estimation by sampling within the structured latent space. Unlike existing methods that focus on maximizing impact while minimizing detectability, especially in scenarios with only power measurements, PI-ExAVAE provides fine-grained control over the magnitude of the impact on the system state. This flexibility allows the model to generate attack vectors that range from subtle changes to significant disruptions, depending on the attacker’s objectives. Section 5 demonstrates this novel capability through case studies where the model adapts to diverse conditions while maintaining physical consistency. As for the voltage amplitude, the reference [27] deduces a simple sufficient condition, namely $\Delta V_{i}\ll V_{i},\forall i$. But this is difficult to use in practice, therefore, reference [27] did not implement this sufficient condition in the blind AC attack, only discussing it when the historical voltage amplitude state quantity is known. However, in the proposed PI-ExAVAE, the adversarial mechanism will naturally learn $\Delta V_{i}\ll V_{i},\forall i$ from the data.

Finally, considering the inherent spatial correlations in power system measurements, we introduce a convolutional neural network to capture these dependencies, making the generated fake data more realistic, physically consistent, and better at bypassing BDD.

So far, based on the description provided in the Section 4, we present the physics-informed PI-ExAVAE network illustrated in Figure 1 and trained with the triple criteria(18)$$L=L_{\mathrm{prior}}+L_{\mathrm{phy}}+L_{\mathrm{adv}}.$$

### 4.3. Training Method and Controllable FDIA Generation

For a given time-point, the composition of the measurement data y (bidirectional line flows and bus injections except for reference bus) is stated as:(19)$$y_{t}=\left[PF_{t}^{ij},PT_{t}^{ij},Pbus_{t}^{i},QF_{t}^{ij},QT_{t}^{ij},Qbus_{t}^{i}\right]$$
where PF, PT, QF, and QT represent real and reactive power injected at the “from” and “to” bus ends, respectively. Pbus and Qbus represent real and reactive power injections at buses. Subscript *i* and *j* are bus numbers. *t* is the time step. Since the number of nodes is usually less than the number of lines (i.e., i<j) within a time step, Pbus and Qbus are padded with 0.

**The strategy for propagating error signals**: According to the observation model y=h(x)+e and the priors over the latent distribution p(z)=N(0,I), the latent space of VAE can be viewed as an embedding of high-dimensional measurements on low-dimensional manifolds.

The encoder maps high-dimensional measurement data y into the low-dimensional latent space z. This process is regularized to ensure that the latent space captures not only the statistical structure of the measurements but also the underlying physical relationships dictated by the AC power flow equations. To achieve this, the encoder loss function includes two components:(20)$$L_{encoder}=L_{\mathrm{prior}}+L_{\mathrm{phy}}.$$
where $L_{\mathrm{prior}}$ ensures that the latent representation z conforms to the prior distribution p(z)=N(0,I), and $L_{\mathrm{phy}}$ enforces that the latent space embedding aligns with the physical laws governing the system states. This formulation ensures that the encoder embeds the high-dimensional measurements into a latent representation that preserves the critical physical relationships necessary for effective FDIA generation.

The decoder reconstructs the manipulated measurements $y_{a}$ from the latent representation z and plays a crucial role in generating physically consistent and stealthy attack vectors. Its loss function comprises two key terms:(21)$$L_{decoder}=L_{\mathrm{phy}}+L_{\mathrm{adv}}.$$
where $L_{\mathrm{phy}}$ guides the decoder to generate attack vectors that adhere to the phase angle in a sufficient condition, ensuring physical consistency with the AC power flow equations, and through adversarial learning, loss $L_{\mathrm{adv}}$ ensures that the reconstructed measurements are indistinguishable from real measurements. It also helps the model learn the voltage amplitude-related sufficient conditions and the power balance equations, both of which are critical for crafting stealthy and effective FDIA. This combination ensures that the decoder not only generates valid attack vectors but also captures the nonlinear relationships between voltage amplitudes and phase angles required for FDIA.

Finally, the discriminator is trained with $L_{\mathrm{adv}}$ described in Equation (Equation 9). Noteworthily, embedding dimension and loss functions are tied to the physical relationships in power systems, ensuring that the PI-ExAVAE can generalize to unseen scenarios. In summary, the overviews of the training procedure are presented in Algorithm 1. The detailed network architectures are listed in Table 2. In the model architecture design, the channel numbers in convolutional layers were determined to balance feature extraction capability and computational cost, while kernel sizes and strides were optimized for the spatial structure of IEEE14 data (Notably, for larger-scale systems, the architecture can be adapted by proportionally increasing the number of convolutional layers and corresponding channels. This approach preserves the hierarchical feature extraction capability while maintaining computational efficiency). Activation functions were selected to enhance non-linearity and training stability. Parameter initialization employed Xavier initialization for weights and zero initialization for biases to ensure balanced gradient flow. The architecture was finalized based on grid search experiments, prioritizing reconstruction accuracy and attack success rate.
**Algorithm 1** Training the PI-ExAVAE attack model.  1:Initialization: $\theta_{\mathrm{Enc}},\theta_{\mathrm{Dec}},\theta_{\mathrm{Dis}}\leftarrow$ network parameters  2:**repeat**  3:   Y← random mini-batch from measurement data y  4:   Z←Enc(Y)  5:   $L_{prior}\leftarrow D_{KL}\left(q\left(Z|Y\right)||p\left(Z\right)\right)$  6:   $\tilde{Y}\leftarrow Dec\left(Z\right)$  7:   $L_{phy}\leftarrow \frac{Y}{∥Y∥}\frac{\tilde{Y}-Y}{∥\tilde{Y}-Y∥}-1$  8:   $L_{adv}\leftarrow log\left(Dis\left(Y\right)\right)+log\left(1-Dis\left(\tilde{Y}\right)\right)$   // Update parameters according to gradients  9:   $\theta_{\mathrm{Enc}}\overset{+}{\leftarrow }-\nabla_{\theta_{\mathrm{Enc}}}\left(L_{prior}+L_{phy}\right)$10:   $\theta_{\mathrm{Dec}}\overset{+}{\leftarrow }-\nabla_{\theta_{\mathrm{Dec}}}\left(L_{phy}-L_{adv}\right)$11:   $\theta_{\mathrm{Dis}}\overset{+}{\leftarrow }-\nabla_{\theta_{\mathrm{Dis}}}L_{adv}$12:**until** maximum epochs (300) reached or loss $L_{adv}$ convergence threshold ($10^{-4}$) met

**Controllable Attack Generation with PI-ExAVAE**: After training the PI-ExAVAE network, two distinct attack generation strategies are available, providing complementary approaches for crafting physically consistent and stealthy FDIA.

Path 1 involves measurement-conditioned generation, where a new measurement y is encoded into the latent space by the encoder and decoded to produce an attack vector $a_{1}$. This path ensures input-specific attacks that align closely with the given measurement while adhering to physical constraints. Path 2 leverages latent sampling-based generation, where a latent vector $\hat{z}$ is randomly sampled from the Gaussian prior p(z)=N(0,I) and decoded to generate an attack vector $a_{2}$. This approach explores a broader operational space and generates attacks beyond the training data, emphasizing the PI-ExAVAE’s extrapolative capability. The flexibility of PI-ExAVAE lies in its ability to interpolate between these two paths, providing controllable FDIA. By linearly interpolating in the latent space,(22)$$z_{inter}=\left(1-val\right)\times Enc\left(y\right)+val\times \hat{z}$$
where val∈[0,1], the PI-ExAVAE generates attacks that range from input-specific (Path 1, val=0) to fully extrapolated (Path 2, val=1). Intermediate values enable a blend of the two, balancing stealth and impact.

Finally, as illustrated in Figure 2, unlike GAN-based models that require two distinct sets of normal measurements for training, PI-ExAVAE significantly reduces data requirements by leveraging physical constraints, allowing it to train with only a single set of normal measurements. Furthermore, the structured latent space enables the generation of attack vectors that deviate significantly from the training data while adhering to physical constraints, a capability that GAN-based methods lack. These features, validated in the subsequent experiments, demonstrate PI-ExAVAE’s controllability and extrapolation capabilities, making it highly adaptable for crafting FDIA under diverse and extreme operational scenarios.

## 5. Case Studies

### 5.1. IEEE14 Test System with NYISO Field Data

As shown in Figure 3 [36], the New York Independent System Operator (NYISO) consists of 11 regions and is marked from A to K, and the IEEE14 system with 11 load buses is chosen as the test system. The software toolbox MATPOWER 7.1 [37] is utilized to generate the measurements, and real power load data in 2023 from NYISO (Rensselaer, NY, USA) are fed into the IEEE14 system. The sampling interval is 5 min. In the experiment, the network information in the standard system is kept unchanged. Each bus and each line is equipped with a power measurement to obtain the active and reactive power of the buses and bidirectional line flows. Therefore, for the snapshot at a given timepoint, the number of measurement data points of the IEEE14 is M=4×L+2×N=106 (recall L=20 is the number of links and N=13 is the number of nodes). The following procedures are utilized to estimate system states using load patterns from NYISO:Step 1. Link the buses of the IEEE14 system to regions of NYISO as follows:$$\left[\begin{array}{ccccccccccc} 2 & 3 & 4 & 5 & 6 & 9 & 10 & 11 & 12 & 13 & 14\\ F & C & I & B & G & K & E & H & J & D & A \end{array}\right]$$
where the row is the bus number of the IEEE14 system and the second row represents the NYISO region index in Figure 3.Step 2. Normalize the load of NYISO to the initial real and reactive load of the corresponding IEEE14 bus, so that the test system operates near the initial state of the IEEE14 system. Due to lack of reactive load information, we assume that the system load has a constant power factor (0.8), so reactive power can be calculated by real power. This assumption can be relaxed if the historical data of reactive power is available.Step 3. Add up the new real power load. Find the ratio of the new total load to the IEEE14 bus initial total load. Multiply this ratio to by generation of all generators.Step 4. Repeat the previous step for reactive power.Step 5. Calculate the system state (x) using AC power flow analysis.Step 6. Calculate the system measurement value y=h(x), where h() is the power flow equation derived from the system structure.Step 7. White Gaussian noise $N(0,0.1^{2})$, i.e., 0.1 p.u. is added to the measurements y.

We construct 11,000 sample data points, of which 1000 are used as the training set for learning the attack model’s parameters, and 10,000 are used as the test set to evaluate its performance on new data. For the IEEE 14-bus system, the PI-ExAVAE is trained using the Adam optimizer with a learning rate of 0.003 and decay rates of 0.5 and 0.999 for the first and second moments, respectively, to ensure stable parameter updates. During training, each batch randomly selects data from the 1000-sample training set and the algorithm continues until either a maximum of 300 epochs is reached or the adversarial loss $L_{adv}$ converges to a specified threshold. Specifically, convergence is determined when the change in $L_{adv}$ across consecutive iterations falls below $10^{-4}$ for 10 consecutive updates, ensuring stability and preventing overfitting. The attack assumes tampering with all measurements at a single time point, while the operator is assumed to have precise knowledge of measurement noise levels for residual-based detection, i.e., the diagonal elements of R in Equation (Equation 1) are set to match the variance of the added noise. This setup evaluates the model’s performance under challenging conditions.

#### 5.1.1. Attack Effectiveness and Controllability Analysis

In order to comprehensively evaluate the effectiveness and controllability of the PI-ExAVAE model proposed in this article for blind FDIA against AC state estimation without voltage measurements, 10,000 independent experiments were ran for each case. We compare our method against three state-of-the-art model-free FDIA methods developed for scenarios without voltage measurements: the self-attention generative adversarial network (SAGAN) proposed in reference [20], the Wasserstein GAN with autoencoder (AE-WGAN) approach presented in reference [21], and the GAN with long short-term memory autoencoder (LSTMAE-GAN) presented in reference [22].

Figure 4 compares the state estimation (SE) residuals for raw measurements (*y*), SAGAN, LSTMAE-GAN, AE-WGAN, and the proposed PI-ExAVAE model under different interpolation values (val). Raw measurements exhibit the highest residuals, often exceeding 100, due to inherent noise causing false positives in residual-based bad data detection (BDD). Among the baseline methods, SAGAN and LSTMAE-GAN reduce residuals but show significant variance and outliers, while AE-WGAN achieves lower residuals with better stability. In contrast, PI-ExAVAE demonstrates superior performance and, most importantly, exceptional controllability. By adjusting the val parameter, PI-ExAVAE progressively reduces residuals, effectively balancing stealth and impact as val transitions from 0 to 1. At higher val values (0.7–1.0), it achieves the lowest median residuals (close to 10) with minimal variance and outliers. These results underscore PI-ExAVAE’s unique ability to dynamically control residuals, offering robust and precise attack generation tailored to specific objectives.

Figure 5 illustrates the detection rates for raw measurements (*y*), LSTMAE-GAN, SAGAN, AE-WGAN, and the proposed PI-ExAVAE model under different val values. A “measurement group” includes all measurements collected at a single time step, including power measurements from multiple nodes. The detection rates are broken down into two components: the light blue base of each bar shows the proportion of time steps with at least one abnormal measurement, while the orange stack represents the total abnormal measurements across all time steps divided by the total number of time steps. Red dots indicate detection rates of 0%.

Raw measurements (*y*) exhibit the highest detection rate, exceeding 25%, primarily due to noise causing high residuals that trigger detection even in the absence of attacks. Among the baseline methods, LSTMAE-GAN shows a relatively high detection rate despite achieving lower residuals in Figure 4. This behavior can be attributed to its focus on capturing temporal correlations between measurements while neglecting the underlying physical relationships described by the power flow equations. As a result, although it reduces residuals, the generated measurements deviate from the physically consistent patterns expected by the system, leading to a higher likelihood of detection. SAGAN reduces the detection rate compared to LSTMAE-GAN but still exhibits a non-negligible level of detected anomalies, reflecting inconsistent stealth. AE-WGAN achieves the lowest detection rate among the baseline methods, with no orange stacks, indicating a complete absence of detected abnormal measurements within groups.

In contrast, PI-ExAVAE demonstrates superior stealth performance, achieving near-zero detection rates for most interpolation values (val). As val increases from 0 to 1, the detection rate of PI-ExAVAE further declines, with red dots indicating a 0% detection rate for val values of 0.5 and beyond. This trend highlights PI-ExAVAE’s ability to adaptively generate attack vectors that reliably evade detection, even as the deviation from the original data increase. The comparison underscores the robustness and controllability of PI-ExAVAE, outperforming all baseline methods in maintaining stealth while effectively bypassing detection mechanisms.

The plot of Figure 6 compares the deviations of estimated ($\left|\right|y_{est}-y{\left|\right|}_{2}$) and attacked measurements ($\left|\right|y_{a}-y{\left|\right|}_{2}$) from the original measurements for varying values of the parameter val (indicated by the color gradient). Each data point represents a measurement group, and the dashed black diagonal line indicates equality ($\left|\right|y_{est}-y{\left|\right|}_{2}$ = $\left|\right|y_{a}-y{\left|\right|}_{2}$). As val increases (from purple to red), the deviations of the attacked measurements from the original increase, highlighting the growing impact of the attack. For higher val values, the points lie above the diagonal, indicating that the attacked measurements deviate more significantly from the original compared to the estimated values. This demonstrates the effectiveness of the attack in creating discrepancies, thereby amplifying its impact on the measurement groups.

Figure 7 illustrates the impact of the proposed PI-ExAVAE model on voltage magnitudes (left) and phase angles (right) across buses in the IEEE-14 system, compared with other methods, including SAGAN, LSTMAE-GAN, and AE-WGAN. The blue dashed line represents the pre-attack state, the orange line corresponds to the SAGAN attack, the pink dotted line represents the LSTMAE-GAN attack, and the red circular markers depict the results of the AE-WGAN attack. Solid lines are used to depict the results of PI-ExAVAE for val values ranging from 0 to 1.0.

The results demonstrate the advanced controllability of PI-ExAVAE, enabling subtle deviations at lower val values and more pronounced, targeted deviations as val increases. At lower val, PI-ExAVAE closely maintains the original voltage profiles, ensuring plausible and realistic patterns. As val approaches 1, it introduces significant deviations in both voltage magnitudes and phase angles, particularly for buses such as Bus 13, effectively amplifying the attack impact while preserving a smooth and structured profile across buses. Notably, the voltage magnitude of the PV bus (Bus 7 in this figure, corresponding to Bus 6 in the standard system) remains unchanged, showcasing its ability to respect physical constraints. In contrast, while methods such as SAGAN, LSTMAE-GAN, and AE-WGAN effectively bypass BDD detection and impact state estimation by generating deviations close to the original state estimation values, they lack the ability to introduce adjustable and structured deviations across multiple buses. In comparison, PI-ExAVAE provides precise control over the magnitude and distribution of post-attack deviations through the parameter val, offering greater flexibility and impact. These comparisons underscore the adaptability of PI-ExAVAE in achieving both realistic and impactful attack patterns.

#### 5.1.2. Analysis of Extrapolative Performance

In order to analyze the extrapolative performance of the proposed PI-ExAVAE, Figure 8 compares the voltage magnitude (left) and voltage angle (right) across the 14 buses of the IEEE-14 system, illustrating the state range covered by the training data (blue shaded region) and the broader state space generated by the physics-guided model (red shaded region). The red boundaries represent the limits of the state space that the physics-guided model is capable of generating, while the blue region reflects the range of states observed within the training data.

The results demonstrate that the physics-guided model is not constrained by the state space observed during training but can extrapolate beyond it, producing a significantly expanded range of physically consistent states. This extrapolative capability is particularly advantageous for generating realistic and effective false measurements in scenarios not explicitly represented in the training data. By leveraging physical constraints and domain knowledge, the model ensures that the generated measurements remain plausible and consistent with the underlying system physics, even under extreme or previously unseen conditions. This ability is critical for constructing robust and stealthy false data injection attacks, as it enables the generation of measurements that are both effective in disrupting state estimation and difficult to detect by traditional monitoring systems.

#### 5.1.3. Performance Analysis Under Different Detectors

To verify the flexibility of the PI-ExAVAE model under various detection scenarios, we evaluated its performance using three detection methods: the largest normalized residual (LNR) test, the Chi-Squared test, and a deep learning-based detector.

**LNR test**: Figure 9 illustrates the detection rates of false measurements under varying thresholds of the normalized maximum residuals (*r*) for the proposed PI-ExAVAE model with different val values, compared to the results for original measurements (*y*), SAGAN, LSTMAE-GAN, and AE-WGAN. The red dashed line (representing original measurements) consistently exhibits high detection rates across all thresholds, exceeding 90% for low thresholds. SAGAN (green dashed line) and LSTMAE-GAN (blue dashed line) reduce detection rates compared to raw measurements but remain significantly less effective than PI-ExAVAE. AE-WGAN (pink dot line) achieves better stealth performance among the baseline methods, but its performance plateaus at higher thresholds. In contrast, PI-ExAVAE demonstrates superior adaptability and stealth performance. By varying the parameter val, PI-ExAVAE achieves progressively lower detection rates as val increases, with near-zero detection rates at higher val values for most thresholds. The solid lines for val = 0.9 and val = 1.0 exhibit the lowest detection rates across all thresholds, highlighting PI-ExAVAE’s ability to evade detection mechanisms effectively. This adaptability makes it possible to balance between stealthiness and impact, outperforming all baseline methods across a wide range of detection thresholds.

**Chi-squared test**: Table 3 presents a quantitative comparison of the success rates for bypassing the $\chi^{2}$-test under 95% and 90% confidence levels for SAGAN, LSTMAE-GAN, AE-WGAN, and the proposed PI-ExAVAE model. Among the baseline methods, AE-WGAN achieves the highest success rates, consistently reaching 100% across both confidence levels, while LSTMAE-GAN performs slightly better than SAGAN, with success rates of 98.6% and 95.3%, respectively, compared to SAGAN’s 95.5% and 90.2%. This slight improvement can be attributed to the lower median residuals observed for LSTMAE-GAN in Figure 4, which reduces its detection probability under the $\chi^{2}$-test.

In contrast, the PI-ExAVAE model consistently achieves 100% success for val=0.5 and val=1.0 across both confidence levels, highlighting its robustness and superior adaptability. Even at val=0, the success rates remain competitive (98.2% at 95% confidence and 96.1% at 90% confidence), outperforming SAGAN and LSTMAE-GAN. These results demonstrate the controllability of PI-ExAVAE, as it effectively tunes val to adapt its attacks and evade various detection mechanisms. The table underscores the superiority of PI-ExAVAE in balancing stealthiness and effectiveness, achieving consistent performance under diverse BDD settings. Compared to baseline methods, PI-ExAVAE provides not only higher success rates but also greater adaptability to real-world scenarios, ensuring robust evasion of detection mechanisms.

**Deep learning-based detector**: In [38], Bhattacharjee et al. proposed a deep latent space clustering (DLSC) framework that combines stacked autoencoders (SAEs) with unsupervised k-means clustering to detect stealthy false data injection attacks (FDIAs) in AC state estimation without requiring labeled data.

Table 4 presents the performance comparison of different attack models under DLSC detection, evaluated using four metrics: Accuracy, Precision, Recall, and F1 Score. These metrics are commonly used to measure the detection system’s ability to identify attacks, with the following interpretations in the context of attack detection:Accuracy: Represents the proportion of correctly classified samples, including both correctly detected attacks and correctly identified normal measurements. A higher accuracy indicates the detection system’s overall reliability in distinguishing between attacks and normal data, while a lower accuracy suggests that the system struggles to identify the true nature of the measurements.Precision: Indicates the proportion of detected attacks that are true attacks (i.e., the accuracy of positive predictions). In this context, higher precision means the detection system generates fewer false alarms, while lower precision implies that many normal measurements are misclassified as attacks.Recall: Reflects the proportion of all true attacks that are successfully detected. Higher recall indicates that the detection system can capture a larger fraction of the actual attacks, while lower recall suggests that many attacks evade detection, indicating better attack stealthiness.F1 Score: Combines precision and recall into a single metric by calculating their harmonic mean. A higher F1 Score represents a good balance between precision and recall, indicating strong detection performance. Conversely, a lower F1 Score highlights that either precision or recall (or both) is compromised, which often correlates with higher attack stealthiness.

In the context of evaluating attack efficacy, higher detection metrics (Accuracy, Precision, Recall, and F1 Score) reflect stronger detection performance and consequently weaker attack efficacy. On the other hand, lower detection metrics suggest that the detection system struggles to identify attacks, demonstrating higher attack stealthiness and success.

The baseline methods—SAGAN, AE-WGAN, and LSTMAE-GAN—utilize their respective generated false measurements for both training and testing, leading to fixed performance levels. Among these methods, SAGAN and LSTMAE-GAN exhibit relatively stronger attack stealthiness, as indicated by lower detection metrics (e.g., F1 Scores of 0.6343 and 0.6739, respectively). In contrast, AE-WGAN achieves higher detection metrics (F1 Score = 0.9009), reflecting weaker attack stealthiness and making it easier for the detection system to identify the attack vectors. These results highlight the limitations of baseline methods in controlling or adapting the attack strength to evade detection effectively.

The proposed PI-ExAVAE model demonstrates a significant advancement by enabling precise control over attack stealthiness through the adjustable parameter val. Trained with false measurements generated at val=0.5, the model achieves perfect evasion (all detection metrics = 1.0000) under testing at the same val. To evaluate the adaptability of PI-ExAVAE, we tested it across a range of val values (from 0 to 1). At val=0, the model achieves an F1 Score of 0.1025, indicating high attack stealthiness as the detection system fails to recognize most attacks. As val increases, detection metrics gradually rise, reflecting a decrease in attack stealthiness. For instance, at val=0.3, the F1 Score reaches 0.9816, and from val=0.5 onwards, all detection metrics reach 1.0000, signifying the attack becomes fully detectable. It is worth noting that the attack vectors generated by PI-ExAVAE at a single intensity (val=0.5) exhibit high consistency with those generated by AE-WGAN, leading to relatively low randomness. This characteristic makes the attacks easier to detect under fixed parameters, as reflected in the recall consistently reaching 1. This limitation is mitigated by the ability of PI-ExAVAE to adaptively adjust attack intensities. Furthermore, introducing a mixture of attack intensities during training could further degrade the performance of the DLSC detection system, amplifying the effectiveness of the proposed method. This ability to systematically adjust attack stealthiness underscores the superior control capability of PI-ExAVAE compared to the fixed-performance baseline methods. Such adaptability makes PI-ExAVAE a versatile tool for generating tailored attacks, offering enhanced stealthiness in diverse scenarios.

#### 5.1.4. Analysis of Model Robustness to Noise

For SCADA systems, measurement errors often lead to deviations between the original measurement data and the power flow data after state estimation, which may affect attacks. However, our PI-ExAVAE model effectively handles noise by leveraging the VAE’s latent space regularization to suppress high-frequency noise and the adversarial training to refine realistic, noise-free outputs. This combination ensures robust noise mitigation while preserving the fidelity of the underlying data structure. In order to verify the robustness of the PI-ExAVAE to noise, three normal noises N(0,0.01), N(0,0.03), and N(0,0.05) are added to the sample data to simulate the measurement error, and the attack success rate of the proposed PI-ExAVAE attack model is calculated and analyzed. Here, we calculate the residual and success rate after the attack under different measuring errors when val=0.5, and the results are shown in Table 5.

As the measuring errors increase, the residual of the false data constructed by the proposed PI-ExAVAE attack model is always smaller than the residual before the attack. When the variance of the measuring error reaches 0.03 and 0.05, the average residuals of the system before the attack increase by 11.09 and 20.36, while those after the attack only increase by 6.36 and 15.7. The results indicate that regardless of the measurement error, the false data generated by the proposed PI-ExAVAE attack model can evade bad data detection. In other words, the proposed attack model exhibits strong robustness to measurement errors.

### 5.2. IEEE118 Test System

The generation of measurements is similar to that of the IEEE14 system in Section 5.1. Specifically, real power load data of the IEEE118 test system is obtained by simulating the pattern of NYISO’s real power load based on the IEEE118 system’s standard load data(23a)$$\begin{array}{cc} m(k) & =m(k-1)+randn\times 0.1 \end{array}$$(23b)$$\begin{array}{cc} PD_{i,j} & =PD_{i,j}^{IE}\times m\left(j\right) \end{array}$$
where x(1)=1, $\left\{k\in Z|1\leq k\leq N_{datas}\right\}$ and randn represents a normally distributed random number. In Equation (23b) PD and IE represent real power load and IEEE118, respectively. *i*, *j* represent the bus number and time point of the real power load. After obtaining the real power load, we use the same steps as described in Section 5.1 to obtain the measurement. Finally, white Gaussian noise N(0,0.01) is added to the measurements. Each bus and each line is equipped with a power measurement to obtain active and reactive power of buses (except for reference bus) and bidirectional line flows. Therefore, for the snapshot at a given timepoint, the measurement data of the IEEE118 includes M=4×L+2×N=978 (recall L=186 is the number of links and N=117 is the number of nodes).

For the IEEE118 system, the specific operations of attacking the model during training are as follows: we perform 300 iterations, and each batch consists of 4000 randomly shuffled samples. In addition, 10,000 samples are for testing, and the model is trained with the same optimizer as in Section 5.1.

Table 6 and Figure 10 collectively demonstrate the flexibility and effectiveness of the proposed PI-ExAVAE model in generating stealthy and impactful false measurements in the IEEE-118 system. Table 6 presents the success rates of the model in bypassing the normalized maximum residuals test with a fixed threshold of $r_{i}=3$. The results show that the PI-ExAVAE model achieves a 100% success rate for val values between 0 and 0.8, with a slight decrease to 95.3% and 90.2% for val=0.9 and val=1.0, respectively. The residual statistics (min, mean, and max) before the attack further highlight the model’s ability to generate effective attacks that reduce the maximum residual below the detection threshold.

Figure 10 complements these findings by illustrating the voltage phasors (magnitude on the left and angle on the right) across 117 buses before and after the attack for different val values. At lower val values, the deviations from the pre-attack states are minimal and stealthy, closely resembling the original measurements, while higher val values introduce more significant and targeted disruptions, particularly for specific buses. Despite the increasing attack intensity, the generated states remain physically plausible and consistent with power system constraints, demonstrating the model’s robustness.

These results highlight the scalability and controllability of the PI-ExAVAE model. By adjusting the parameter val, the model can effectively balance stealthiness and impact, adapting to complex detection scenarios such as the normalized maximum residuals test. Its ability to maintain physical feasibility while scaling attack intensity underscores its suitability for large-scale systems and diverse detection mechanisms.

## 6. Conclusions

This study introduces the PI-ExAVAE model, a physics-guided generative framework for creating stealthy and impactful false measurements in power systems. By integrating domain knowledge and physical constraints, the model generates measurements that remain consistent with system physics while effectively evading detection mechanisms, such as $\chi^{2}$-tests and normalized maximum residuals tests. Unlike SAGAN, which produces abrupt and less structured deviations, the PI-ExAVAE model offers fine-grained control through the parameter val, enabling a balance between stealthiness and impact and adaptability to varying detection thresholds. The model demonstrates strong extrapolative capabilities, generating physically plausible states beyond the training data, and excels in large-scale systems like IEEE-118, maintaining effectiveness and scalability. Experimental results confirm the model’s superior performance over SAGAN, achieving near-perfect detection evasion across diverse scenarios.

In summary, PI-ExAVAE sets a new standard for generating stealthy and adaptable false measurements in power systems, making it a robust tool for analyzing system vulnerabilities and advancing the study of smart grid security. 

## Figures and Tables

**Figure 1 sensors-25-00943-f001:**
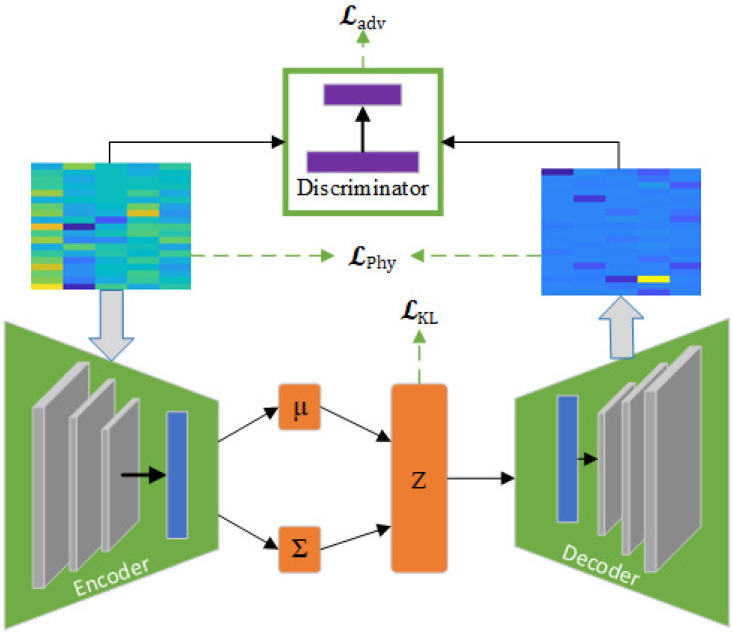
The structure of physics-informed extrapolative adversarial VAE.

**Figure 2 sensors-25-00943-f002:**
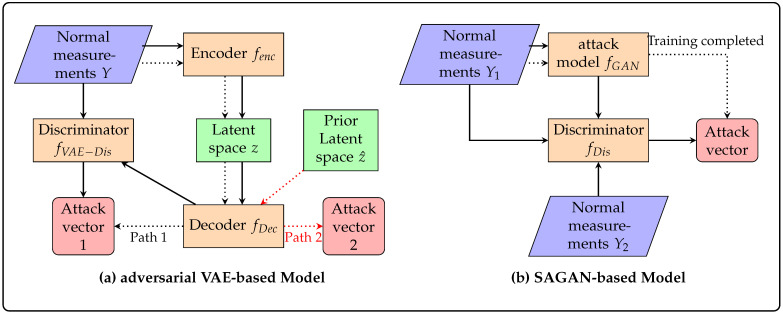
The difference between attack methods based on GAN [20] and our PI-ExAVAE. The dotted line indicates the vector generation path after training is completed.

**Figure 3 sensors-25-00943-f003:**
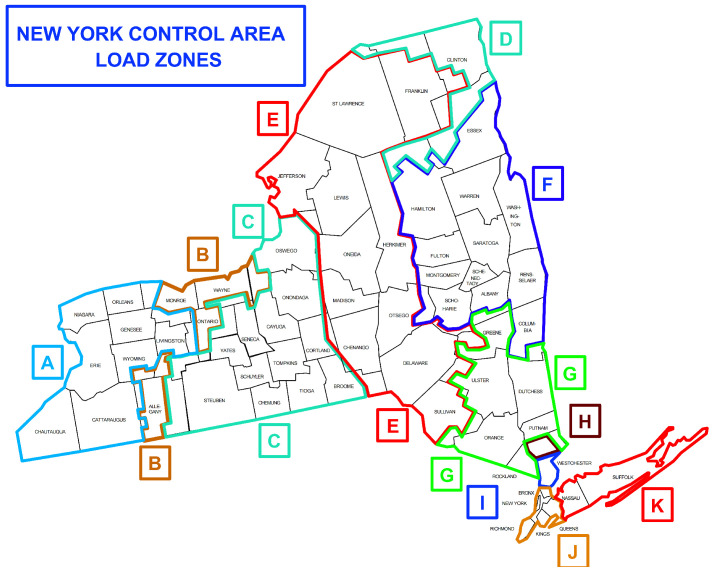
Region index of NYISO.

**Figure 4 sensors-25-00943-f004:**
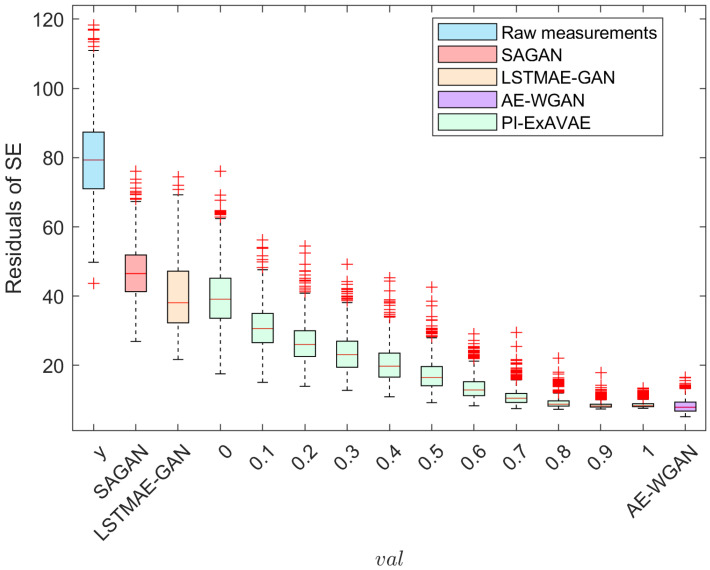
The residuals of the PI-ExAVAE under different val are compared with those of the raw measurements (*y*), LSTMAE-GAN, SAGAN, and AE-WGAN methods.

**Figure 5 sensors-25-00943-f005:**
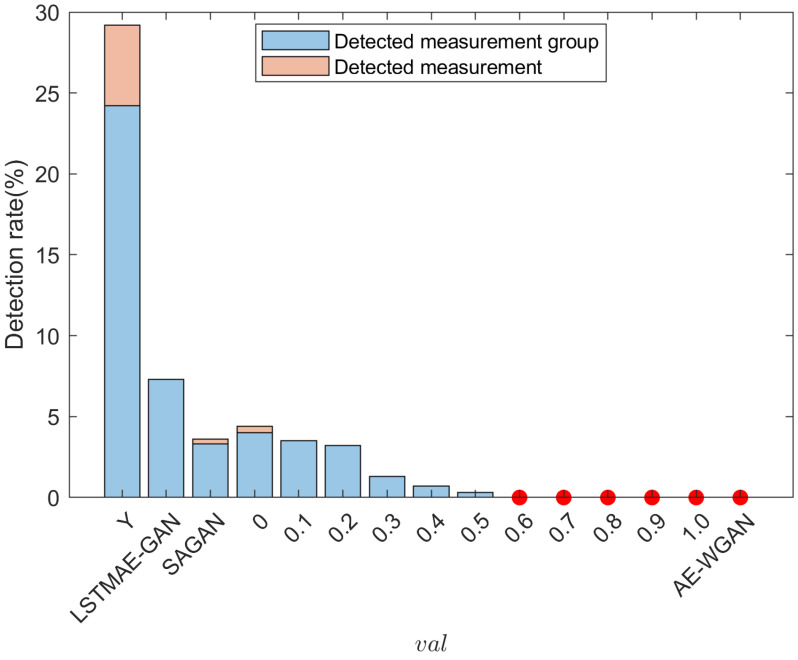
The detection rate of the PI-ExAVAE under different val are compared with those of the raw measurements (y), LSTMAE-GAN, SAGAN, and AE-WGAN. A “measurement group” includes all measurements collected at a single time step, including power measurements from multiple nodes. The light blue base of each bar shows the proportion of time steps with at least one abnormal measurement, while the orange stack represents the total abnormal measurements across all time steps divided by the total number of time steps. Red dots indicate detection rates of 0%.

**Figure 6 sensors-25-00943-f006:**
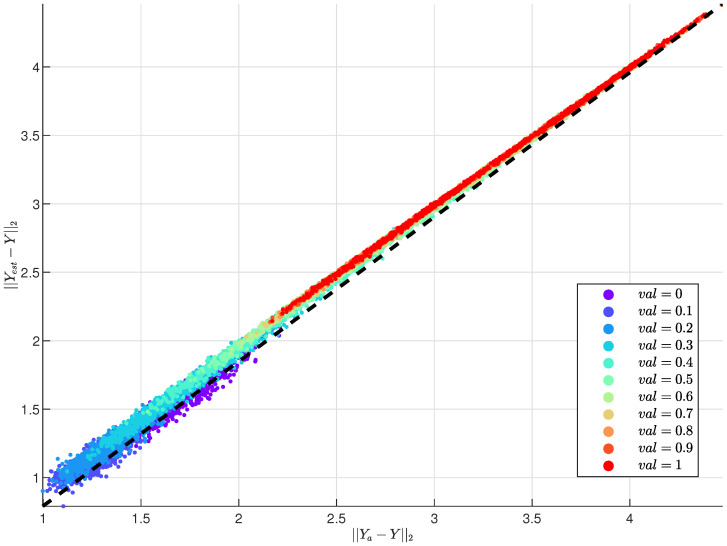
Deviations of estimated and attacked measurements (with different val) compared with the original.

**Figure 7 sensors-25-00943-f007:**
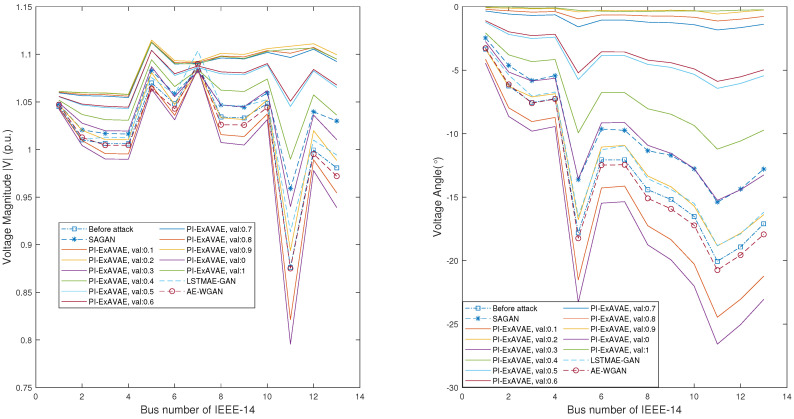
Voltage phasors of 13 buses (except for reference bus) before and after the attack with different methods and different val. The left figure shows the voltage magnitude, while the right figure shows the voltage phase angle. PI-ExAVAE demonstrates precise control over state estimation deviations, allowing for gradual adjustments in magnitude and phase angle as val varies.

**Figure 8 sensors-25-00943-f008:**
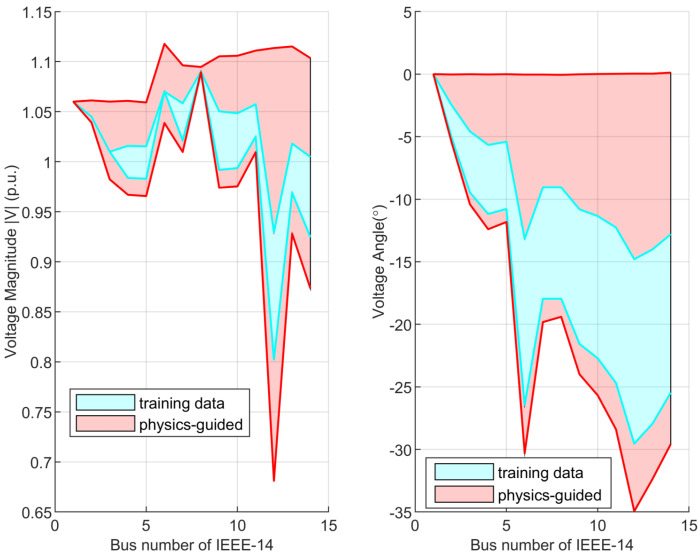
Comparison of voltage magnitude (**left**) and voltage angle (**right**) between training data and the physics-guided model for IEEE-14 buses.

**Figure 9 sensors-25-00943-f009:**
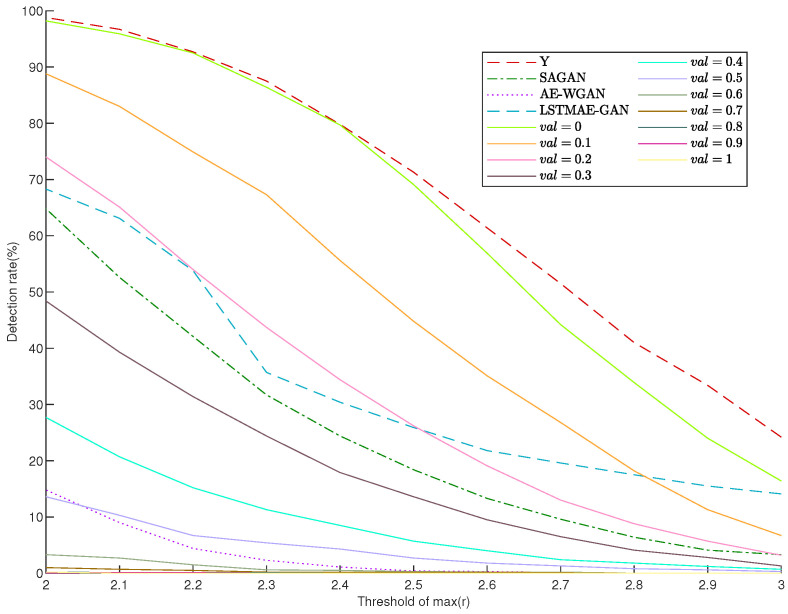
Detection rates under different thresholds of the normalized maximum residual r for various val values in the proposed PI-ExAVAE model compared to original measurements, SAGAN, LSTMAE-GAN, and AE-WGAN.

**Figure 10 sensors-25-00943-f010:**
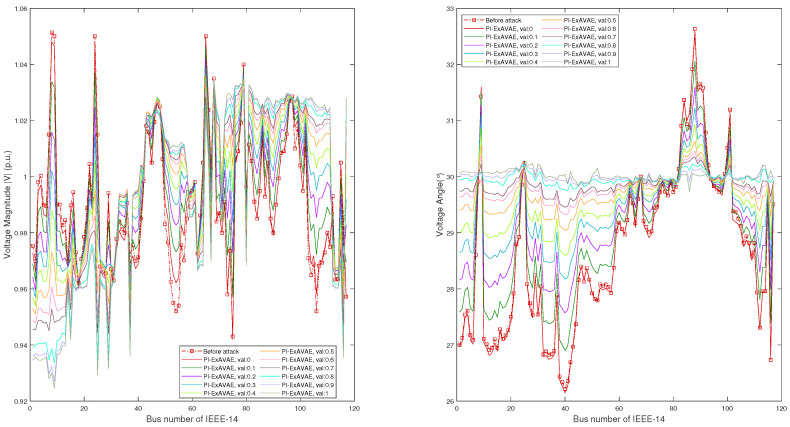
Voltage phasors of 117 buses before and after the attack. The left figure shows the voltage magnitude, while the right figure shows the voltage phase angle.

**Table 1 sensors-25-00943-t001:** Comparison of different FDIA methods.

Method	Key Features	Voltage Required	Controllable SE Deviations
Afrin et al. [26]	Sneaky-FGSM selectively perturbs high-variance measurements using multi-layer perceptron (MLP) surrogate models	Yes	No
Costilla et al. [21]	AE-WGAN: autoencoder serving as a fixed surrogate estimator	Yes	No
Narang et al. [22]	LSTMAE-GAN: LSTMAE embeds temporal dependencies within the autoencoder-based surrogate estimator	Yes	No
Jiao et al. [20]	SA-GAN leverages the self-attention mechanism to effectively capture long-range dependencies in power measurement data	No	No
**PI-ExAVAE**	A VAE guided by physical priors to control post-attack state estimation deviations via latent space controllability	No	Yes

**Table 2 sensors-25-00943-t002:** Architectures of the PI-ExAVAE attack model for IEEE14.

Encoder	Decoder	Discriminator
Conv2d(1, 8, (3, 2), (2, 1)), ReLU	ConvTranspose2d(hidden, 32, (2, 2), (1, 1)), ReLU	Conv2d(1, 16, (3, 2), (2, 2))
Conv2d(8, 16, (3, 2), (2, 1)), ReLU	ConvTranspose2d(32, 16, (3, 2), (1, 1)), ReLU	SpectralNorm, LeakyReLU
Conv2d(16, 32, (3, 2), (1, 1)), ReLU	ConvTranspose2d(16, 8, (3, 2), (2, 1)), ReLU	Conv2d(16, 32, (3, 2),(2, 1))
Conv2d(32, 64, (2, 2), (1, 1)), ReLU	ConvTranspose2d(8, 1,(4, 3), (2, 1)), ReLU	SpectralNorm, LeakyReLU
FC(hidden, latent)-Mean	Tanhshrink	Conv2d(32, 64, (3, 2), (2, 1))
FC(hidden, latent)-Logvar		SpectralNorm, LeakyReLU
		AdaptiveAvgPool2d(1)
		Linear(64, 1), Sigmoid

**Table 3 sensors-25-00943-t003:** Comparison of passing the $\chi^{2}$-test with SAGAN, LSTMAE-GAN, and AE-WGAN.

Confidence Level		SAGAN	LSTMAE-GAN	AE-WGAN	PI-ExAVAE		
					val **= 0**	**val = 0.5**	**val = 1**
Success Rate (%)	95%	95.5	98.6	100	98.2	100	100
	90%	90.2	95.3	100	96.1	100	100

**Table 4 sensors-25-00943-t004:** Performance comparison of different attack models under DLSC detection.

Method		Accuracy	Precision	Recall	F1 Score
SAGAN		0.4645	0.4816	0.9290	0.6343
AE-WGAN		0.8900	0.8197	1.0000	0.9009
LSTMAE-GAN		0.5160	0.5081	1.0000	0.6739
PI-ExAVAE	val=0	0.5270	1.0000	0.0540	0.1025
	val=0.1	0.7445	1.0000	0.4890	0.6568
	val=0.2	0.8945	1.0000	0.8890	0.9411
	val=0.3	0.9502	1.0000	0.9643	0.9816
	val=0.4	1.0000	1.0000	1.0000	1.0000
	val=0.5	1.0000	1.0000	1.0000	1.0000
	val=0.6	1.0000	1.0000	1.0000	1.0000
	val=0.7	1.0000	1.0000	1.0000	1.0000
	val=0.8	1.0000	1.0000	1.0000	1.0000
	val=0.9	1.0000	1.0000	1.0000	1.0000
	val=1	1.0000	1.0000	1.0000	1.0000

**Table 5 sensors-25-00943-t005:** The success rate of IEEE14 system under different noise levels.

Normal Noise	Residual Before Attack	Residual After Attack	Success Rate
	Min/Mean/Max	Min/Mean/Max	
N(0,0.01)	43.70/79.55/118.26	10.00/17.08/35.95	100%
N(0,0.03)	54.84/90.64/146.24	15.14/23.44/65.68	99.7%
N(0,0.05)	63.03/99.91/175.64	25.42/32.78/90.20	99.5%

**Table 6 sensors-25-00943-t006:** The success rates of IEEE118 system.

		Min/Mean/Max	Success Rate
Residual before attack		643.07/742.75/863.78	90%
PI-ExAVAE	val=0	83.82/94.58/121.52	100%
	val=0.1	83.50/89.86/106.81	100%
	val=0.2	101.38/126.85/172.20	100%
	val=0.3	152.74/191.51/229.13	100%
	val=0.4	221.85/272.18/318.69	100%
	val=0.5	310.69/361.57/412.27	100%
	val=0.6	399.59/452.75/504.51	100%
	val=0.7	489.46/542.29/599.57	100%
	val=0.8	571.53/622.95/683.75	100%
	val=0.9	638.06/693.08/738.09	95.3%
	val=1.0	640.67/739.92/840.78	90.2%

## Data Availability

Data are available upon request.

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
