# Peer review of "Controllable Blind AC FDIA via Physics-Informed Extrapolative AVAE"

_sensors, 2025, doi:10.3390/s25030943_

Round 1

Reviewer 1 Report

Comments and Suggestions for Authors

The work is well written methodically. The content is presented clearly. The presented models, algorithms are well described. I did not find any errors. In its current form, this manuscript can be published.

Reviewer 2 Report

Comments and Suggestions for Authors

The manuscript is about applying a physical law in conjunction with a variational encoder and a discriminator neural network to the generation of false data that can fool the state estimator of a power grid.

There are many problems with the manuscript:

Abstract: "priors" -- if the authors mean prior distribution, then the authors should clarify prior distribution of what. Readers need to be informed and not left guessing.

Page 2: Regarding this statement "Several works adopted AC state estimation to build the FDIA [12–14]", this is a gross understatement because research in the area dates back to more than a decade ago. Much has been published since the appearance of ref [1] in 2011 (the conference version appeared in 2009).

Page 2: Instead of cramming a literature review into the introduction, please create a separate "Related Work" section following good academic tradition.

Some notable related papers haven't been cited, e.g., the following paper also uses an autoencoder but for attack detection:

@ARTICLE{9927371,
  author={Bhattacharjee, Arnab and Mondal, Arnab Kumar and Verma, Ashu and Mishra, Sukumar and Saha, Tapan K.},
  journal={IEEE Transactions on Smart Grid},
  title={Deep Latent Space Clustering for Detection of Stealthy False Data Injection Attacks Against AC State Estimation in Power Systems},
  year={2023}, volume={14}, number={3}, pages={2338-2351}, doi={10.1109/TSG.2022.3216625}}

Another example: the following paper uses an autoencoder and a GAN for false data injection, so this is highly relevant to the authors:

@article{narang2024physical,
author = {Jagendra Kumar Narang and Baidyanath Bag},
title = {Physical model learning based false data injection attack on power system state estimation},
journal = {Sustainable Energy, Grids and Networks},
volume = {40}, pages = {101524}, year = {2024}, issn = {2352-4677},
doi = {10.1016/j.segan.2024.101524}}

There are many more related papers that haven't been cited; the authors should really do their homework.

Page 2: The discussion of related work jumps from [12] to [20] back to [11] with seemingly no pattern. In the newly created "Related Work" section, please discuss the literature systematically rather than bouncing from a random paper to another. Observe the pattern, extract commonality, differences and trend, please.

Page 2: "Generative extrapolation" -- does "generative" capability not imply "extrapolation" capability?

Page 3: Eq. (1) is missing a subscript denoting the optimization variable.

Page 3: The definition of matrix R doesn't look right.

Page 4: Why use different symbols -- p and q -- to denote probability distribution?

Page 4: Notation in Eq. (6) is incorrect. Enc(y) outputs a vector, which is a sample of the population following distribution q(z|y). Enc(y) is not q(z|y) itself. Same goes with Dec(z) and p(y|z).

Page 4: Citation needed for Eq. (7).

Page 5: This sentence seems to be incomplete: "...the sufficient conditions for bus active and reactive power injection attacks to be "invisible" are the same as the active and reactive power of the relevant line flow."

Page 5: Citation needed for Eq. (12).

Page 5: Derivation of Eq. (14) needs to be clarified.

Page 5: Symbols for vectors y and a need to be explicitly defined.

Page 6: What are the "targeted sampling strategies"?

Page 6: What is the evidence that the "unique capability" is unique?

I cannot tell what the SAGAN method is but I'm not convinced that an unknown method is used for comparison when there are already so many competing attack methods out there; recall the uncited references mentioned earlier.

Furthermore, the chi-square test is for bad data detection, and not a defense, as ref [1] has already pointed out more than a decade ago. Please implement a recognized, recently proposed defense and use that to test the proposed attack.

Problems that are less serious but need to be fixed:

Page 1: "preventive strategies" --> "mitigation strategies". If we are dealing with false data injection, we are no longer preventing false data injection, but mitigating the impact of false data injection.

Page 4: If subscript "llike" stands for log likelihood, then please clarify to the readers.

Page 6: "literature [18]" --> "reference [18]". "Literature" is a body of works/references; please note the difference.

Page 6: "triple criterion" --> "triple criteria"

In terms of math formatting, matrices are conventionally formatted as bold upright capitalized. It's also beneficial to apply different formatting to vectors and scalars. Eq. X --> Eq. (X).

If this submission receives a Major Revision, please annotate/label/tag every change with "Reviewer X Comment Y". I will reject any revision without these annotations/labels/tags.

Reviewer 3 Report

Comments and Suggestions for Authors

Clarify own innovation and impact.

Is solution in Fig 1 own innovation? If so, what is new ? How is it better ?

Explain validity and generalisability of results, such as  in Fig 4,5,6,7,8, 9, 10; Table 2, 3.

From computing point of view, the paper should explain in good technical details on computing and space complexity of the computing solutions, such as compare own solution in Algorithm 1  with existing relevant solutions.

The paper should explain from computing perspective how the solutions select parameters and how initial values of parameters were assigned, such as in Table 1.

The paper should discuss more relevant recent work, such as:

Image denoising with generative adversarial networks and its application to cell image enhancement

IEEE Access 8, 82819-82831

Rational choice of methods and choice of evaluation criteria.

About advantages of own solution, provide formal proof.

Reviewer 4 Report

Comments and Suggestions for Authors

1. The method used in this paper has been widely studied and is common in the past literature. The author lacks comparison and analysis of the proposed method with existing methods, which is inappropriate.

2. How are the relevant parameters given by Eq. 11 obtained? Some comments are recommended.

3. How is the noise in this article obtained and processed? It is inappropriate to have no explanation and analysis about the noise part.

Comments on the Quality of English Language

The English could be improved

Round 2

Reviewer 2 Report

Comments and Suggestions for Authors

The manuscript has been significantly improved, especially by including more comparisons in Sec. 5 and implementation of the DLSC detector, but there are still problems:

-- Previously there were 14 mentions of "blind". This time, there are 17 mentions, but there's still no clear definition of "blind" in the context of FDIA.

-- While Response 4 explains the structure of the Sec. 2 "Related Work", the first sentence of each paragraph should explain what the paragraph is about, instead of making the readers guess. The readers do not have access to the responses letter.

-- Lines 100-105: Reference [22] is summarised here, but a comparison with the authors' scheme, which also uses autoencoders, is expected. A literature review should not be a list of mini summarises. There should be insights on the strengths and weaknesses of each scheme. Comparisons are expected. A table for comparing different schemes is a common strategy.

-- Line 118: Citation needed for "Ornstein-Uhlenbeck process and regression theorem".

-- I do not agree with Response 5, but I'm not spending more time on this debate.

-- As part of Response 13, there's now a derivation of Eq. (14), but the sign of the term $V_iV_jB_{ij}\sin(\hat{\theta}_{ij})$ seems to be inconsistent. The correctness of the manuscript needs more attention. I don't have time to check other equations, so I hope the authors can proofread again.

-- In Algorithm 1, how is "deadline" quantified? Please be specific.

-- What are "measurement groups"? I don't know how to interpret "measurement group" in Figure 5.

-- Lines 459-460: I think it's unproductive to vaguely criticise other schemes to promote the authors' own scheme. For example, from Figure 7, I don't know how to interpret "SAGAN's abrupt and less structured deviations", "LSTMAE-GAN's inconsistent bus-level control", "AE-WGAN's lack of fine-grained scalability". These adjectives seem irrelevant to how we evaluate attacks.

-- Figure 7: Text is not legible.

-- How does Figure 7 help the readers evaluate the effectiveness and stealthiness of attacks?

Comments on the Quality of English Language

Line 92-94: "For example, Jiao et al. [20] employed a self-attention-GAN-based (SAGAN) technique—though the latter may lack flexibility for real-time adjustments." -- What does "latter" refer to? The long hyphen is not appropriate here.

Line 100: [22] --> First author's surname et al. [22].

Reviewer 4 Report

Comments and Suggestions for Authors

I have no comments

Author Response

Thank you for taking the time to review our work. We appreciate your effort and are grateful for your feedback throughout the process.

Round 3

Reviewer 2 Report

Comments and Suggestions for Authors

I appreciate the authors' perseverance and cooperation. The manuscript has been significantly improved. With the improvements, I consider this manuscript to be of better quality than most papers in the Sensors journal.

Just minor correction: reference entry [29] is incomplete.